# An active site loop toggles between conformations to control antibiotic hydrolysis and inhibition potency for CTX-M β-lactamase drug-resistance enzymes

Shuo Lu[1], Liya Hu [2], Hanfeng Lin [1], Allison Judge [2], Paola Rivera [1], Murugesan Palaniappan[3], Banumathi Sankaran [4], Jin Wang [1], B. V. Venkataram Prasad [2] & Timothy Palzkill [1,2] ✉

β-lactamases inactivate β-lactam antibiotics leading to drug resistance. Consequently, inhibitors of β-lactamases can combat this resistance, and the β-lactamase inhibitory protein (BLIP) is a naturally occurring inhibitor. The widespread CTX-M-14 and CTX-M-15 β-lactamases have an 83% sequence identity. In this study, we show that BLIP weakly inhibits CTX-M-14 but potently inhibits CTX-M-15. The structure of the BLIP/CTX-M-15 complex reveals that binding is associated with a conformational change of an active site loop of β-lactamase. Surprisingly, the loop structure in the complex is similar to that in a drug-resistant variant (N106S) of CTX-M-14. We hypothesized that the pre-established favorable loop conformation of the N106S mutant would facilitate binding. The N106S substitution results in a ~100- and 10-fold increase in BLIP inhibition potency for CTX-M-14 and CTX-M-15, respectively. Thus, this indicates that an active site loop in β-lactamase toggles between conformations that control antibiotic hydrolysis and inhibitor susceptibility. These findings highlight the role of accessible active site conformations in controlling enzyme activity and inhibitor susceptibility as well as the influence of mutations in selectively stabilizing discrete conformations.

Bacterial resistance to antibiotics is increasing and thereby reducing treatment options for antimicrobial therapy[1]. β-lactam antibiotics, including penicillins, cephalosporins, and carbapenems, act by inhibiting transpeptidase enzymes involved in bacterial cell wall biosynthesis. β-lactams are heavily prescribed, representing approximately 65% of antibiotic usage worldwide[2]. Resistance to β-lactam antibiotics is most commonly due to bacterial production of β-lactamases. These enzymes inactivate the drugs by catalyzing the hydrolysis of the β-lactam ring. Further, the genes encoding β-lactamases are often plasmid-encoded, allowing horizontal transfer so that β-lactamases are widespread among Gram-positive and Gram-negative bacteria[3,4].

There are four classes of β-lactamases (A, B, C, and D) that are grouped according to amino acid sequence homology[5]. Among these, the class A β-lactamases are widespread and are an important source of resistance[3]. The canonical class A β-lactamase, TEM-1, is common in Gram-negative bacteria and rapidly hydrolyzes penicillins and many cephalosporins, but not extended-spectrum cephalosporins such as

[1]Department of Pharmacology and Chemical Biology, Baylor College of Medicine, Houston, TX, USA. [2]Verna and Marrs McLean Department of Biochemistry and Molecular Biology, Baylor College of Medicine, Houston, TX, USA. [3]Center for Drug Discovery, Department of Pathology & Immunology, Baylor College of Medicine, Houston, TX 77030, USA. [4]Department of Molecular Biophysics and Integrated Bioimaging, Berkeley Center for Structural Biology, Lawrence Berkeley National Laboratory, Berkeley, CA, USA. ✉e-mail: timothyp@bcm.edu

cefotaxime[6]. In response to the selective pressure of extended-spectrum cephalosporin use, hundreds of variants of β-lactamases have emerged that exhibit increased hydrolysis of these drugs[6]. These include the emergence of CTX-M β-lactamases, which are ~ 40% identical to TEM-1 in sequence and rapidly hydrolyze the drugs[7–9]. CTX-M enzymes are now a major source of clinical resistance to extended-spectrum cephalosporins among Gram-negative bacteria[4].

CTX-M enzymes are divided into five subgroups based on amino acid sequence homology, including CTX-M-1, CTX-M-2, CTX-M-8, CTX-M-9, and CTX-M-25; with the names based on prominent members of the subgroups[7–10]. There is >10% sequence divergence between subgroups but enzymes within a subgroup differ by only a few amino acids. The CTX-M-14 enzyme from the CTX-M-9 subgroup and, particularly, the CTX-M-15 enzyme from the CTX-M-1 subgroup are the most widespread among Gram-negative pathogens[8,9] (Supplementary Figure S1).

Although the CTX-M enzymes that initially emerged readily hydrolyze extended-spectrum cephalosporins such as cefotaxime, they poorly hydrolyze ceftazidime. However, CTX-M variants have evolved that have gained the ability to hydrolyze ceftazidime[4,6]. These variants commonly contain active site substitutions such as P167S and D240G, which each increase $k_{cat}/K_M$ values for ceftazidime hydrolysis by > 10-fold[11,12]. In addition, the N106S substitution is commonly found among CTX-M variants[8]. We have previously shown that this substitution leads to decreased catalytic activity against cefotaxime and ceftazidime but resistance emerges because of a large increase in the thermodynamic stability and expression levels of the variant[13]. Asn106 is not in the active site but rather is a second-shell residue. The N106S substitution results in a conformational change of the active site 103-106 loop such that key interactions between the loop residues Asn104 and Tyr105 and cephalosporin substrates are altered, thereby reducing activity[13]. Interestingly, position 106 is naturally a serine in the TEM-1 enzyme, which does not hydrolyze extended-spectrum cephalosporins[13].

The spread of class A enzymes such as TEM-1 and CTX-M has led to the development of small-molecule inhibitors of these β-lactamases, including clavulanic acid, sulbactam, tazobactam, and avibactam[14–16]. The inhibitors are used in combination with a β-lactam drug and they restore efficacy by inactivating β-lactamase. Resistance to inhibitors is on the rise, however, and new classes of these molecules are needed[17].

β-lactamase inhibitory protein (BLIP) is produced by *Streptomyces clavuligerus* and inhibits class A β-lactamases with potencies ranging from μM to high pM[18–20]. For example, it inhibits TEM-1 β-lactamase with a $K_i$ of 0.5 nM but has much weaker potency for the CTX-M-14 enzyme, with a $K_i$ of 330 nM[18,21,22]. Structural studies of BLIP alone and in complex with the TEM-1, SHV-1, and KPC-2 β-lactamases reveal it is a two-domain protein with a concave binding surface[18,23–25]. The two domains exhibit pseudosymmetry with 29% sequence identity between the domains, suggesting the protein evolved from a tandem duplication[26]. BLIP binds a protruding loop on class A enzymes and inhibits the enzymes by extending a loop from each domain of BLIP into the active site of β-lactamase[23] (Fig. 1a).

Here we show that BLIP potently inhibits CTX-M-15 β-lactamase with a $K_i$ of 2.9 nM despite displaying such a weak inhibition of the related enzyme, CTX-M-14 ($K_i$ = 330 nM). Further, the structure of CTX-M-15 in complex with BLIP shows that the 103-106 loop of the enzyme is in an altered conformation compared to the apo-CTX-M-15 structure. Upon BLIP interacting with the 103-106 loop, the hydrogen bond network of Asn106 observed in the apo enzyme is disrupted and the peptide backbone between Asn104 and Tyr105 is flipped 180°. The side chain of Tyr105 is also in a different rotamer conformation that is reminiscent of the conformation of the 103-106 loop seen in the BLIP complex with TEM-1 β-lactamase. Interestingly, the apo CTX-M-14 N106S mutant structure shows a conformation of the 103-106 loop that is similar to that observed in TEM-1 and CTX-M-15 complexes with BLIP[13,23]. We reasoned that since the N106S substitution stabilizes this altered loop conformation in the CTX-M-14 apo-enzyme, it may

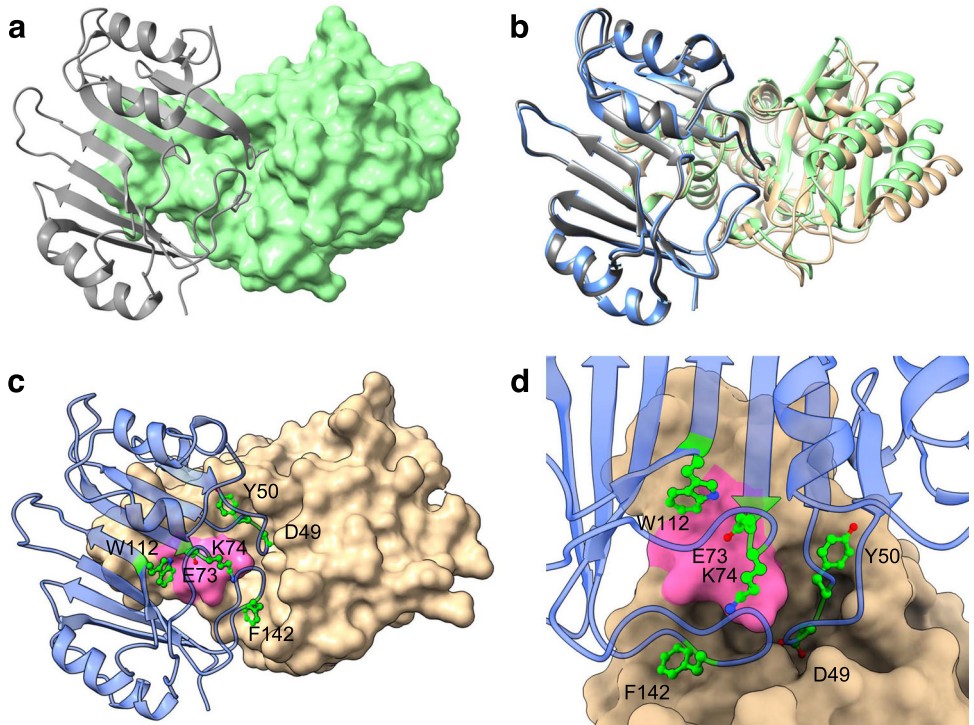

**Fig. 1 | Structure representation of BLIP binding to TEM-1 and CTX-M-15 β-lactamases. a** Ribbon diagram of BLIP (gray) binding to TEM-1 β-lactamase (green, surface) (PDB id: 1JTG). **b** Structure of BLIP (gray) binding to TEM-1 β-lactamase (green) superimposed with BLIP (blue) binding to CTX-M-15 β-lactamase (tan). **c** Structure of BLIP (blue) binding to CTX-M-15 β-lactamase (tan). Key BLIP residues that interact with CTX-M-15 are shown in ball and stick (green) and labeled. The 103–106 loop region of CTX-M-15 is highlighted in pink. **d** BLIP/CTX-M-15 active site region rotated clockwise 90° from view in **c**.

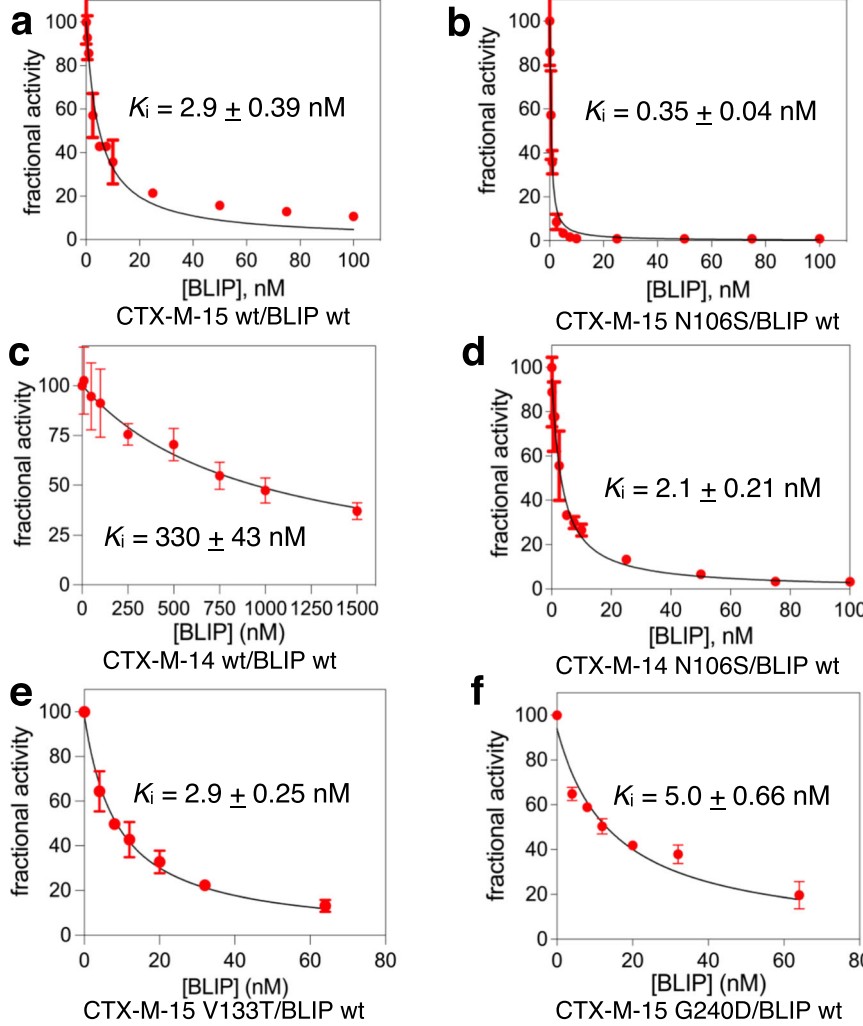

**Fig. 2 | Determination of inhibition constants for β-lactamases.** Inhibition constants ($K_i$) for BLIP with **a** CTX-M-15 wild type, **b** CTX-M-15 N106S, **c** CTX-M-14 wild type, **d** CTX-M-14 N106S, **e** CTX-M-15 V133T, and **f** CTX-M-15 G240D β-lactamases. BLIP concentrations are indicated on the X-axis and the initial velocity of nitrocefin hydrolysis in the presence of BLIP divided by the velocity in the absence of BLIP is shown on Y-axis. The initial velocities shown are the mean of at least two determinations with the associated error bar indicating the standard deviation of the value. Values for $K_i$ were determined by fitting to the Morrison equation (Methods). The error on the $K_i$ value is the standard error based on the fitting to the equation. Source data are provided as Source data file.

enhance the binding of BLIP to this enzyme. In fact, the N106S substitution in both CTX-M-14 and CTX-M-15 results in much stronger BLIP inhibition (10-100 fold) of these enzymes. These results indicate that BLIP potency is sensitive to the conformation of the 103-106 loop in the apo enzyme such that the conformation of the 103-106 active site loop controls CTX-M enzyme activity towards cefotaxime as well as susceptibility to BLIP inhibition. In wild-type CTX-M enzymes with Asn106, the loop is in a conformation consistent with rapid cefotaxime hydrolysis and is inhibited weakly by BLIP, while with Ser106, the loop toggles to a TEM-like structure associated with lower cefotaximase activity and more potent inhibition by BLIP. Finally, the structure of CTX-M-15 with BLIP showing the TEM-like loop structure suggests both loop conformations can be sampled by CTX-M enzymes even with Asn106.

## Results

### BLIP is a potent inhibitor of CTX-M-15 but not the related CTX-M-14 β-lactamase

Previous studies have shown that BLIP inhibits class A β-lactamases with a wide range of potencies, from μM to high pM[18,20]. The structural features and energetics of BLIP inhibition of the CTX-M class enzymes can provide an additional framework for new inhibitor design. Since CTX-M-15 is the most prevalent CTX-M enzyme among Gram-negative pathogens[3,4], we tested its susceptibility to inhibition by BLIP. We cloned the CTX-M-15 gene into a protein expression plasmid and purified the enzyme after overexpression in *E. coli* (Materials and Methods). The steady-state kinetic parameters for hydrolysis of the colorimetric β-lactam substrate nitrocefin were determined, revealing a $k_{cat}$ of 420 s$^{-1}$, $K_M$ of 41 μM, and $k_{cat}/K_M$ of 10.2 μM$^{-1}$s$^{-1}$. These values are consistent with previously determined kinetic parameters[27]. BLIP potently inhibits the CTX-M-15 enzyme with a $K_i$ of 2.9 nM (Fig. 2). This is similar to the $K_i$'s observed with several other class A enzymes including TEM-1[21,23,28], KPC-2[25,29], SME-1[20], and *B. anthracis* Bla1[20]. It is, however, surprisingly more potent than that previously observed for BLIP inhibition of the related (83% amino acid sequence identity) CTX-M-14 β-lactamase ($K_i$ ~800 nM) (Supplementary Fig. 1)[22]. We confirmed the previous findings[22] that BLIP is a much weaker inhibitor of CTX-M-14, albeit with a somewhat lower $K_i$ (330 nM) (Fig. 2).

### CTX-M-15 exhibits a conformational change of an active site loop when bound with BLIP

To investigate the interactions driving the tight binding of BLIP to CTX-M-15 β-lactamase, we screened BLIP/CTX-M-15 mixtures to produce suitable crystals and determined the X-ray crystal structure of the

**Table 1 | Crystallographic statistics for CTX-M-15/BLIP X-ray structure**

| | CTX-M-15/BLIP |
|---|---|
| **Data collection** | |
| Space group | I 1 2 1 |
| Cell dimensions | |
| *a, b, c* (Å) | 47.48, 81.86, 129.12 |
| α, β, γ (°) | 90.00 95.94 90.00 |
| Resolution (Å) | 34.51–1.40 (1.42–1.40) |
| *R*merge | 0.075 (0.354) |
| *I/σI* | 14.6 (5.0) |
| Completeness (%) | 98.37 (97.10) |
| Redundancy | 7.0 (6.9) |
| **Refinement** | |
| Resolution (Å) | 33.49–1.40 (1.45–1.40) |
| No. reflections | 94919 (9334) |
| *R*work | 0.1512 (0.1673) |
| *R*free | 0.1664 (0.1845) |
| No. atoms | |
| Protein | 3194 |
| Ligand/ion | 0 |
| Water | 770 |
| *B*-factors (Å$^2$) | |
| Average | 15.27 |
| Protein | 12.18 |
| Water | 28.08 |
| R.m.s. deviations | |
| Bond lengths (Å) | 0.006 |
| Bond angles (°) | 0.95 |

Values in parentheses are for the highest-resolution shell.

complex. The BLIP/CTX-M-15 complex was determined at 1.4 Å resolution in space group I 1 2 1 (Table 1). As expected, BLIP binds the protruding loop region of β-lactamase, and the BLIP Asp49 and Phe142-containing loops are inserted into the β-lactamase active site, consistent with inhibition of the enzyme (Fig. 1b, c, Supplementary Fig. 2). The overall binding orientation of BLIP and β-lactamase is similar to that observed for the BLIP/TEM-1 complex with BLIP (PDB id:1JTG) and the protruding domain of β-lactamase from both structures is superimposable (Fig. 1b).

The BLIP loops that insert into the active site of β-lactamase play a key role in the inhibition of the enzyme[21,23] (Fig. 1a). Asp49 is present on the His45-Tyr51 loop that inserts into the active site and its carboxylate group mimics the carboxyl group that is present in β-lactam antibiotics[23] (Figs. 1, 3, Supplementary Fig. 3). In the TEM-1/BLIP structure, the Asp49 carboxylate OD1 forms hydrogen bonds with the Ser235 Oγ and Arg244 NH1, while the Asp49 carboxylate OD2 hydrogen bonds to Ser130 Oγ and Ser235 Oγ (Fig. 3d). In contrast, in the CTX-M-15/BLIP structure, Asp49 is in an altered conformation where the carboxylate OD1 forms a hydrogen bond with Ser235 Oγ while OD2 participates in hydrogen bonds with Ser 70 Oγ and Ser237 Oγ (Fig. 3c). If Asp49 were to assume an identical rotamer conformation as that observed in the TEM-1 complex, the OD1 oxygen would clash with Ser235 Oγ (Fig. 3c, d). In addition, BLIP Asp49 hydrogen bonds to Ser237 Oγ in the CTX-M-15 structure but this bond is not possible in TEM-1, as this residue is Ala237.

BLIP residue Phe142 is on the 136-145 loop and makes interactions in the active site in both the TEM-1/BLIP and CTX-M-15/BLIP structures. In the TEM-1 structure, Phe142 interacts with Tyr105, Asn170, Ala237, Gly238, and Glu240[23] (Fig. 3f). Phe142 in the CTX-M-15 structure is in an

altered rotamer conformation where it interacts extensively with the β-sheet and omega loop residue Thr171 but loses interactions with Tyr105 (Fig. 3e). In addition, the hydrogen bonds between TEM-1 Glu104 and the mainchain N of BLIP Tyr143 are not present in the CTX-M-15/BLIP complex because position 104 is Asn.

The structure of BLIP in complex with CTX-M-15 β-lactamase shows extensive interactions with the protruding loop region of β-lactamase. Within this region are β-lactamase residues 103-106, which form a 4-residue β-turn with residues Asn104 and Tyr105 localized in the active site (Fig. 4b, Supplementary Fig. 2, 4, Table 2). The 103−106 residue region meets the β-turn criteria of the *i* and *i* + 3 Cα atoms being separated by less than 7 Å distance[30–32] (Table 2). However, it lacks a hydrogen bond between the NH and CO of residues *i* and *i* + 3 and the phi-psi angles of *i* + 1 and *i* + 2 do not match those of classical type I, I', II, and II' turns[30]. The PROMOTIF structural analysis program classifies residues 103-106 as a Type IV β-turn in CTX-M-14 and a Type VIII turn in CTX-M-15[33]. However, the phi-psi angles are similar for both enzymes for *i* + 1 and *i* + 2 residues 104-105 and the turn falls just outside of the Type VIII angles for CTX-M-14 (Table 2). In these structures, rather than a hydrogen bond between the main chain NH and CO of residues *i* and *i* + 3, the mainchain O and N of Val103 (*i*) form hydrogen bonds with the Asn106 (*i* + 3) side chain Nδ and Oδ atoms, respectively (Fig. 4a, b). TEM-1 residues 103-106 also form a Type IV β-turn but the presence of Ser106 rather than Asn106 alters the hydrogen bonding pattern where Ser106 forms hydrogen bonds with the mainchain O of Val103 and NH of Thr133 (Fig. 4d). This alternate pattern is coincident with the peptide bond between residues 104 and 105 being flipped ~180° relative to that in CTX-M enzymes such that the residue 104 main chain O is hydrogen bonded to the side chain of Asn132 (Fig. 4d, Table 2).

A comparison of the structure of the CTX-M-15/BLIP complex with that of the CTX-M-15 β-lactamase apo-enzyme reveals conformation changes that occur in the 103-106 β-turn of the enzyme upon BLIP binding (Fig. 5). As noted, in the CTX-M-15 apo enzyme the Asn106 Nδ and Oδ atoms hydrogen bond with the mainchain O and N, respectively, of Val103 (Fig. 4b). Also, the Asn104 Oδ hydrogen bonds with Asn132 Nδ and the mainchain O of Tyr105 bonds with the Asn132 mainchain N. In the BLIP complex with CTX-M-15, Asn106 is shifted out of position to hydrogen bond with the mainchain N and O of Val103 and the dihedral angles around the peptide bond between Asn104 and Tyr105 are flipped so that the residue 104 main chain O hydrogen bonds with Asn132, as observed in the TEM-1 apo-enzyme (Figs. 3e, 4d, e). Coincident with these changes, the side chain of Tyr105 assumes a different rotamer conformation where the phenoxy group is angled down toward the active site (Figs. 4e, 5).

Interestingly, the altered conformation of the 103−106 loop of CTX-M-15 in complex with BLIP is similar to that of TEM-1 in complex with BLIP (Fig. 4e, f) (Supplementary Fig. 2a, b; Supplementary Fig. 4). In particular, the peptide bond between Asn104 and Tyr105 is in the same orientation and the side chain of Tyr105 is also shifted to a similar position in the TEM-1/BLIP structure[23]. In this position, the side chain hydroxyl of Tyr105 makes a hydrogen bond with the BLIP mainchain N of Gly48 and hydrophobic interactions with Ala47 in both the BLIP/TEM and BLIP/CTX-M-15 structures. The flip in the orientation of the peptide bond between residues 104-105 in CTX-M-15 also facilitates a hydrogen bonding interaction between BLIP Glu73 and the mainchain N of Tyr105 that is equivalent to the interaction in the BLIP/TEM-1 structure (Fig. 3g, h). In addition, a structural alignment of the CTX-M-14 and CTX-M-15 apo-enzymes with the BLIP/CTX-M-15 structure shows that, in the absence of the conformational change of the 103-106 turn, Trp112 of BLIP would clash with the main chain O of Val103 and the side chain Nδ of Asn106 for both CTX-M-14 and CTX-M-15 (Fig. 6, Supplementary Fig. 2). Also, alignment of the CTX-M-14 N106S apo-enzyme with the BLIP/CTX-M-15 structure shows the substitution and associated conformational change relieves the steric clash with BLIP

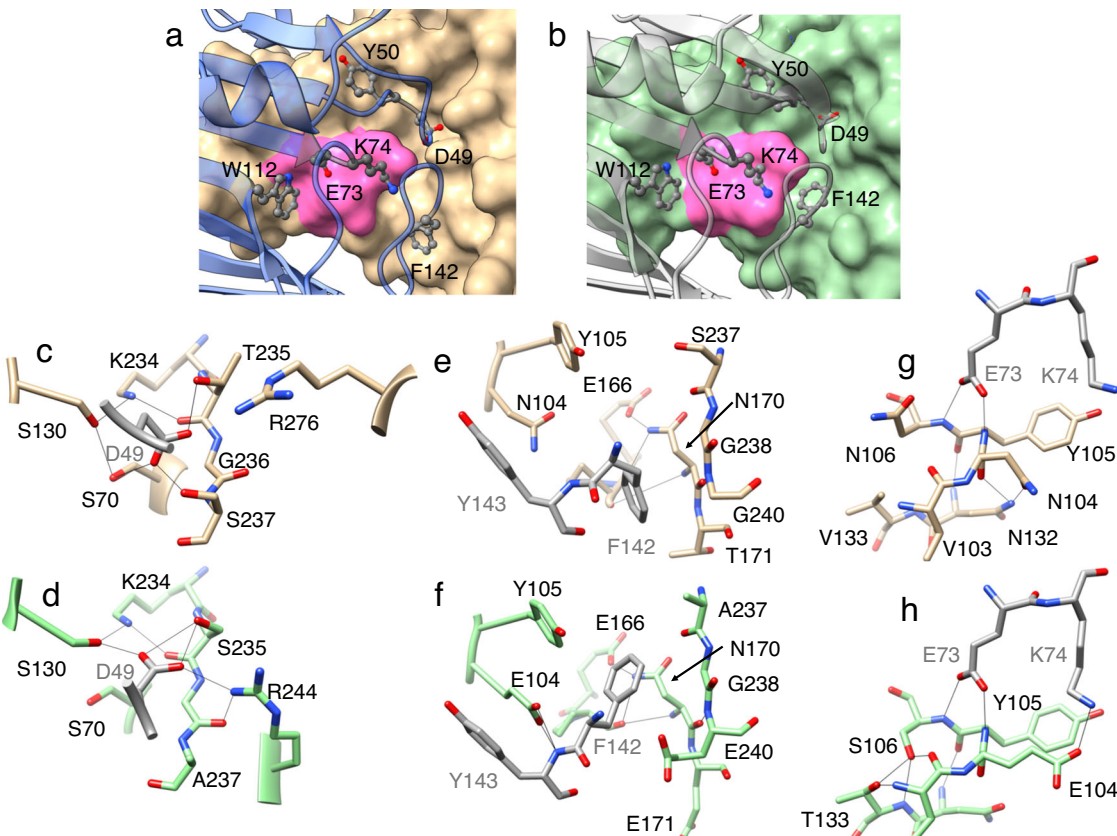

**Fig. 3 | Structure diagrams of BLIP bound to CTX-M-15 and TEM-1 β-lactamases.**
**a** Active site region of CTX-M-15 β-lactamase (tan, surface representation), in complex with BLIP (blue, ribbon). Key BLIP residues are shown in gray ball-and-stick. Oxygen is shown in red and nitrogen in blue. The CTX-M-15 103-VNYN-106 loop is highlighted in pink. **b** Active site region of TEM-1 β-lactamase (green, surface), in complex with BLIP (PDB id: 1JTG). Key BLIP residues are shown in gray ball-and-stick. The TEM-1 103-VEYS-106 loop is highlighted in pink. Panels **a** and **b** are shown in the same orientation as panels **c**-**h**. **c** BLIP Asp49 (gray) residue interactions with the active site of CTX-M-15 β-lactamase (tan). Oxygen is shown in red and nitrogen in blue. Hydrogen bonds are shown as thin black lines as predicted by UCSF Chimera. **d** BLIP Asp49 (gray) interactions with the active site of TEM-1 β-lactamase (green)(PDB id: 1JTG). **e** BLIP Phe142-Tyr143 (gray) interactions with CTX-M-15 (tan). **f** BLIP Phe142-Tyr143 (gray) interactions with TEM-1 (green). **g** Structure of the CTX-M-15 Val103-Asn106 loop showing the hydrogen bond network (tan). The interactions of the region with BLIP residues Glu73 and Lys74 are also shown. **h** Structure of the TEM-1 Val103-Asn106 loop showing hydrogen bond network (green). The interactions of the region with BLIP residues Glu73 and Lys74 are also shown.

Trp112, consistent with the tight binding of BLIP and this mutant (Fig. 6e). Alanine mutagenesis studies of BLIP have shown that Trp112 is a hotspot residue for binding class A β-lactamases in that the BLIP W112A mutant shows >10-fold loss in inhibitor potency towards all β-lactamases tested[20]. Taken together, these results suggest the conformational change of the CTX-M-15 103-106 loop is essential to accommodate tighter binding interactions with BLIP.

Examination of the CTX-M-15 apo-enzyme structure versus the structure in complex with BLIP also reveals that the Gly213-Ser220 loop in the helix 10 region undergoes a conformational change (Fig. 5, Supplementary Fig. 5). In particular, the dihedral angles around the peptide bond between Thr216 and Gly217 are flipped in the CTX-M-15 apo enzyme relative to the complex with BLIP (Supplementary Fig. 5). The change is associated with a movement of Thr215 mainchain O and the side chain of Thr216, which forms a hydrogen bond with the mainchain O of Asp49 in the BLIP 45-52 loop (Supplementary Fig. 5). This interaction could help stabilize the BLIP 45-52 loop in position to make interactions with the CTX-M-15 active site residues.

### Conformational alterations of the CTX-M 103-106 loop enhances BLIP inhibition potency

The conformation of the 103-106 β-turn in the CTX-M-14 and CTX-M-15 apo-enzymes is identical (Fig. 4a, b). In previous studies, we showed that the naturally occurring N106S substitution, when introduced into CTX-M-14 β-lactamase, results in a change in the conformation of the 103–106 turn where the 104–105 peptide bond flips ~180° (Fig. 4c, Supplementary Fig. 2). These changes result in an increase in the thermodynamic stability of the CTX-M-14 enzyme but a decrease in catalytic efficiency for hydrolysis of the extended-spectrum cephalosporin, cefotaxime[13]. The changes due to the N106S mutation result in a conformation of the 103-106 turn in CTX-M-14 that closely matches that observed for TEM-1 β-lactamase (Fig. 4c, d, Table 2). In addition, the conformation of the 103–106 turn in the CTX-M-14 N106S mutant is similar to that observed in the CTX-M-15/BLIP and TEM-1/BLIP complexes (Fig. 4c, e, f, Supplementary Fig. 2).

BLIP is a weak inhibitor of CTX-M-14 β-lactamase and a strong inhibitor for CTX-M-15, despite the high sequence identity between the enzymes. We reasoned that because the N106S mutation results in a conformation of the 103–106 turn that is close to that in the TEM-1 and CTX-M-15 β-lactamase/BLIP complexes, BLIP would display more potent inhibition of the CTX-M-14 N106S mutant enzyme compared to the wild-type CTX-M-14. This proved to be the case as BLIP inhibits the CTX-M-14 N106S enzyme with a $K_i$ of 2.1 nM compared to a $K_i$ of 330 nM (160-fold) for wild-type CTX-M-14[22]. We also tested the inhibition potency of CTX-M-15 β-lactamase containing the N106S substitution. We hypothesized that, because there is a shift in the conformation of the CTX-M-15 103-106 loop in the apo-enzyme versus the BLIP complex (Fig. 4), the N106S substitution would stabilize the conformation containing the altered peptide bond between Asn104 and Tyr105 and enhance binding affinity for BLIP. Consistent with this

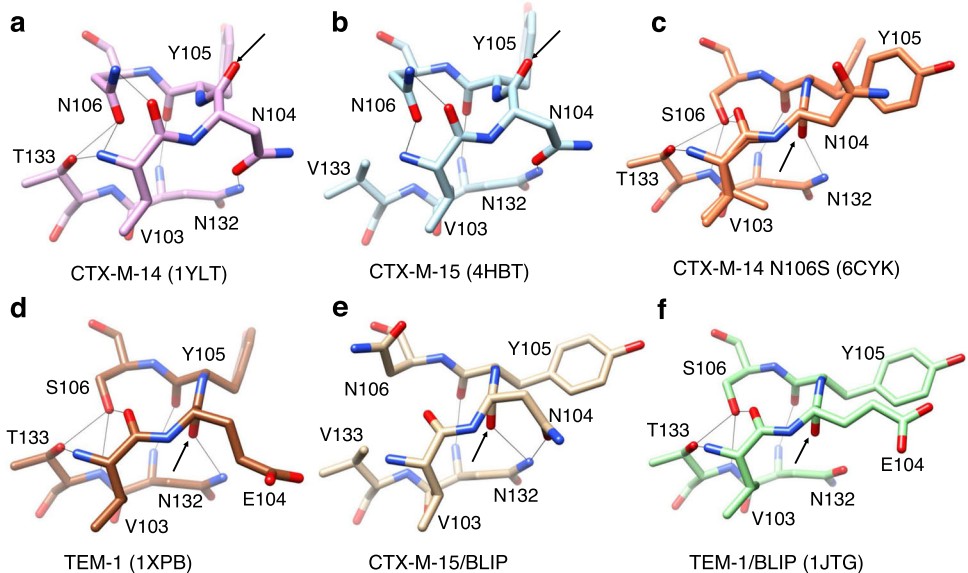

**Fig. 4 | Structures of the 103-106 loop and interacting residues for CTX-M-14, CTX-M-14 N106S, CTX-M-15, TEM-1 apo-enzyme, and CTX-M-15 and TEM-1 in complex with BLIP.** a CTX-M-14 V103-N106 loop structure (pink). b CTX-M-15 V103-N106 loop structure (blue). c CTX-M-14 N106S mutant V103-S106 loop structure (orange). d TEM-1 V103-S106 loop structure (brown). e CTX-M-15 V103-N106 loop structure from the complex with BLIP (tan). f TEM-1 V103-S106 loop structure from a complex with BLIP (green). Oxygen and nitrogen atoms are shown in red and blue, respectively. Hydrogen bonds are shown as thin black lines as predicted by UCSF Chimera. The black arrow in each panel points to the position of the β-lactamase residue 104 carbonyl oxygen, which indicates the orientation of the 104–105 peptide bond. Note the flip in the peptide bond orientation between panels **a** and **b** versus **c**–**f**.

hypothesis, introducing the N106S mutation into CTX-M-15 causes the $K_i$ for BLIP inhibition to drop from 2.9 nM to 0.35 nM (8.5 fold) (Fig. 2). Taken together, these results indicate that the conformation of the 103-106 loop is an important structural determinant of BLIP binding

## Table 2 | Dihedral angles of β-lactamase residues 103–106

| β-lactamase | dihedral angles(°) | | | $C_{\alpha1}$-$C_{\alpha4}$ distance (Å) |
|---|---|---|---|---|
| | omega | phi | psi | |
| CTX-M-14 V103 | 166.9 | −110.8 | −139.7 | 6.71 |
| CTX-M-14 N104 | 193.1 | −90.0 | −13.4 | |
| CTX-M-14 Y105 | 184.3 | −156.3 | 99.4 | |
| CTX-M-14 N106 | 191.9 | −131.8 | 55.1 | |
| CTX-M-15 V103 | 177.7 | −125.4 | −140.8 | 6.59 |
| CTX-M-15 N104 | 187.5 | −88.4 | −14.9 | |
| CTX-M-15 Y105 | 186.7 | −156.2 | 102.0 | |
| CTX-M-15 N106 | 194.6 | −133.2 | 47.7 | |
| CTX-M-15 V103/BLIP | 171.9 | −134.0 | 169.3 | 6.62 |
| CTX-M-15 N104/BLIP | 179.4 | −64.1 | 157.5 | |
| CTX-M-15 Y105/BLIP | 185.2 | 64.3 | 62.9 | |
| CTX-M-15 N106/BLIP | 180.4 | −123.6 | 49.6 | |
| TEM-1 V103 | 174.2 | −113.0 | 174.7 | 6.49 |
| TEM-1 E104 | 181.4 | −53.5 | 143.1 | |
| TEM-1 Y105 | 179.6 | 58.7 | 73.0 | |
| TEM-1 S106 | 176.9 | −129.9 | 60.8 | |
| TEM-1 V103/BLIP | 177.2 | −131.8 | 167.8 | 6.43 |
| TEM-1 E104/BLIP | 177.8 | −47.7 | 139.2 | |
| TEM-1 Y105/BLIP | 180.2 | 67.2 | 64.2 | |
| TEM-1 S106/BLIP | 180.0 | −131.7 | 57.9 | |
| CTX-M-14 N106S- V103 | 176.5 | −105.0 | 113.9 | 6.34 |
| CTX-M-14 N106S- N104 | 187.2 | −56.7 | 140.3 | |
| CTX-M-14 N106S- Y105 | 178.4 | 58.7 | 65.8 | |
| CTX-M-14 N106S- S106 | 177.1 | −36.0 | 66.6 | |

and inhibition of class A β-lactamases and that mutation of a residue outside of the active site, N106S, stabilizes a loop conformation that enhances BLIP binding but reduces enzyme catalytic efficiency for cefotaxime. In effect, the N106S substitution converts CTX-M-14 and -15 from a CTX-M-like enzyme to a TEM-like enzyme by toggling the conformation of the 103-106 loop.

A remaining question is why does BLIP inhibit CTX-M-15 β-lactamase with much higher potency than CTX-M-14? We first examined amino acid differences between CTX-M-14 and -15 that are localized near the active site. In CTX-M-14 β-lactamase, residue 133 is Thr, while for CTX-M-15 it is Val (Fig. 4a, b). The side chain of Thr133 of CTX-M-14 makes a hydrogen bond with Asn106, while Val133 in CTX-M-15 cannot make the equivalent interaction. This could alter the stability of the loop and thus the occupancy of the altered loop conformation and inhibition potency (Fig. 4a, b). We introduced the V133T substitution into CTX-M-15 and tested BLIP inhibition of the mutant enzyme. However, the $K_i$ for BLIP inhibition of the V133T mutant was identical to that of wild-type CTX-M-15, indicating this difference does not account for differences in potency (Fig. 2e). We also examined position 240 on the β-sheet forming a wall of the active site, which is Asp in CTX-M-14 and Gly in CTX-M-15. However, the introduction of the G240D substitution into CTX-M-15 resulted in no change in BLIP potency ($K_i$ = 5 nM)(Fig. 2f). Therefore, the most obvious active site differences between CTX-M-14 and CTX-M-15 do not account for the observed difference in BLIP potency.

The difference in potency of BLIP inhibition of CTX-M-14 versus CTX-M-15 could also lie in relative stability/dynamics differences in the apo enzymes or differences in the energies of the BLIP complexes. As noted, the 103–106 active site loop displays an identical conformation in both the CTX-M-14 and CTX-M-15 apo-enzymes (Fig. 4a, b). If the loop conformation in the apo-enzyme affects BLIP binding, and the loop adopts a similar structure in the BLIP complex, the enzymes would be expected to exhibit similar susceptibility to inhibition, yet they differ in BLIP $K_i$ by 160-fold. One possibility is that the CTX-M-15 103–106 loop is less stable than that in CTX-M-14 and that it samples other conformations, including the TEM-like conformation seen in the CTX-M-15/BLIP structure (Fig. 4e). Consistent with this hypothesis, we

measured the thermal stability of the CTX-M-14 and -15 enzymes using differential scanning fluorimetry (DSF) and found that CTX-M-15 is less stable (49.6°) than CTX-M-14 (51.2°) (Supplementary Fig. 6).

To further assess differences in 103–106 loop dynamics, we performed 6 replicates of 100 ns molecular dynamics simulations of the CTX-M-14 (PDB id:1YLT), CTX-M-14 N106S (6CYK) and CTX-M-15 (4HBT) apo-enzymes. We focused on 100 ns in order to provide multiple replicates (6X) rather than a single longer run. From these replicates, we observe consistent differences between the CTX-M enzymes tested. Of the three enzymes, the CTX-M-14 conformation was the

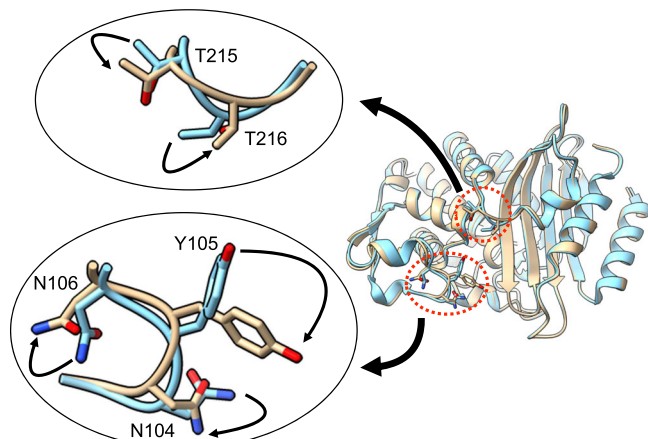

**Fig. 5 | Conformational changes in CTX-M-15 β-lactamase associated with binding of BLIP.** The ribbon diagram at right shows a structural alignment of CTX-M-15 apo-enzyme (blue) (PDB id: 4HBT) with the CTX-M-15 enzyme from the binding complex with BLIP (tan) (BLIP not shown for clarity). Conformational changes are seen in the BLIP-bound CTX-M-15 enzyme in the Val103-Asn106 turn region of the protruding loop as well as the Gly113-Ser220 loop containing Thr215 and Thr216. The insets show the conformational changes with the arrows showing the direction of change from CTX-M-15 apo to BLIP-bound CTX-M-15.

most stable, with little variation over the simulation as indicated by the low RMSD for residues 103–106 with an average value for all simulations of 1.15 Å and a standard deviation (SD) of 0.45 Å (Fig. 7a). The CTX-M-14 N106S loop was more flexible with and average RMSD of 1.58 Å and SD of 1.12 Å (Fig. 7b). In contrast, residues 103–106 showed the highest RMSD for the CTX-M-15 apo enzyme, with an average RMSD of 2.13 Å and SD of 1.15 Å (Fig. 7c). These results are consistent with the 103–106 loop of the CTX-M-15 apo-enzyme sampling more conformations than that in CTX-M-14. Also, consistent with the CTX-M-15 103–105 region having high mobility, the normalized B-factors for all residues in the 103–106 loop, but particularly Tyr105 and Asn106, are significantly higher for CTX-M-15 compared to CTX-M-14 (Table 3). The B-factors were normalized using the equation $(B - <B>)/\sigma$, where $<B>$ is the average B-factor for main chain and side chains for the structure and $\sigma$ is the SD[34]. Therefore, low values in Table 3 indicate residue rigidity and high values indicate flexibility. Further, negative values in Table 3 indicate a residue more rigid than average for a structure. Thus, the CTX-M-14 103–106 loop is more rigid than that in CTX-M-15 based on B-factors. Also, as noted, the CTX-M-15 enzyme shows lower thermal stability than CTX-M-14, consistent with the molecular dynamics results. Taken in aggregate, these findings are consistent with the hypothesis that CTX-M-15 samples more conformations of the 103–106 loop, including ones that facilitate BLIP binding. A further implication is that the more stable loop conformation in CTX-M-14 provides a higher barrier to BLIP binding since the conformation must change for BLIP binding. Note, however, that although CTX-M-15 showed more conformational heterogeneity in the simulation, the flip of the peptide bond between Asn104 and Tyr105 was not observed during the simulation for the CTX-M-15 apo enzyme. Thus, although the simulation results are consistent with the hypothesis that the CTX-M-15 loop samples more conformations, they did not identify the specific loop conformation associated with tight BLIP binding.

We also performed three replicate 100 ns simulations of the CTX-M-15/BLIP structure determined here and models of the CTX-M-14/BLIP and CTX-M-14 N106S/BLIP structures. These structures were

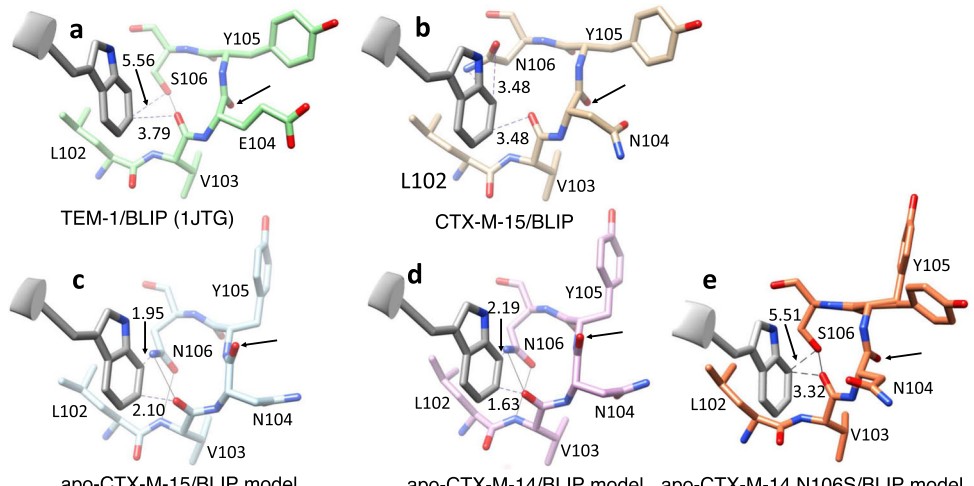

**Fig. 6 | Structures and structural alignments of the interaction of BLIP Trp112 with the TEM-1, CTX-M-15, and CTX-M-14 β-lactamases. a** Structure of BLIP (gray) in complex with TEM-1 β-lactamase (green) showing the interaction of BLIP Trp112 with the β-lactamase 103–106 loop region (PDB id:1JTG). **b** Structure of CTX-M-15 (tan) in complex with BLIP showing the BLIP Trp112 interaction with the 103–106 loop. **c** Structure alignment of CTX-M-15 apo-enzyme (blue) (PDB id:4HBT) with CTX-M-15 in the CTX-M-15/BLIP structure. CTX-M-15 from the BLIP structure is not shown for clarity. **d** Structure alignment of CTX-M-14 apo-enzyme (pink) (PDB id:1YLT) with CTX-M-15 in the CTX-M-15/BLIP structure. CTX-M-15 β-lactamase from the BLIP structure is not shown for clarity. **e** Structure alignment of CTX-M-14

N106S apo-enzyme (orange) (PDB id:6CYK) with CTX-M-15 in the CTX-M-15/BLIP structure. CTX-M-15 β-lactamase from the BLIP structure is not shown for clarity. In all panels, hydrogen bonds are shown as continuous black lines while contact distances are shown as dashed blue lines and labeled with the distance. Note that, in the absence of the conformational change of the 103-106 loop in the CTX-M-14 and CTX-M-15 apo-enzymes, there are close contacts between the loop and BLIP Trp112. Also, the N106S substitution in CTX-M-14 N106S (6CYK) relieves the steric clash. The black arrow in each panel points to the position of the β-lactamase residue 104 carbonyl oxygen, which indicates the orientation of the 104-105 peptide bond.

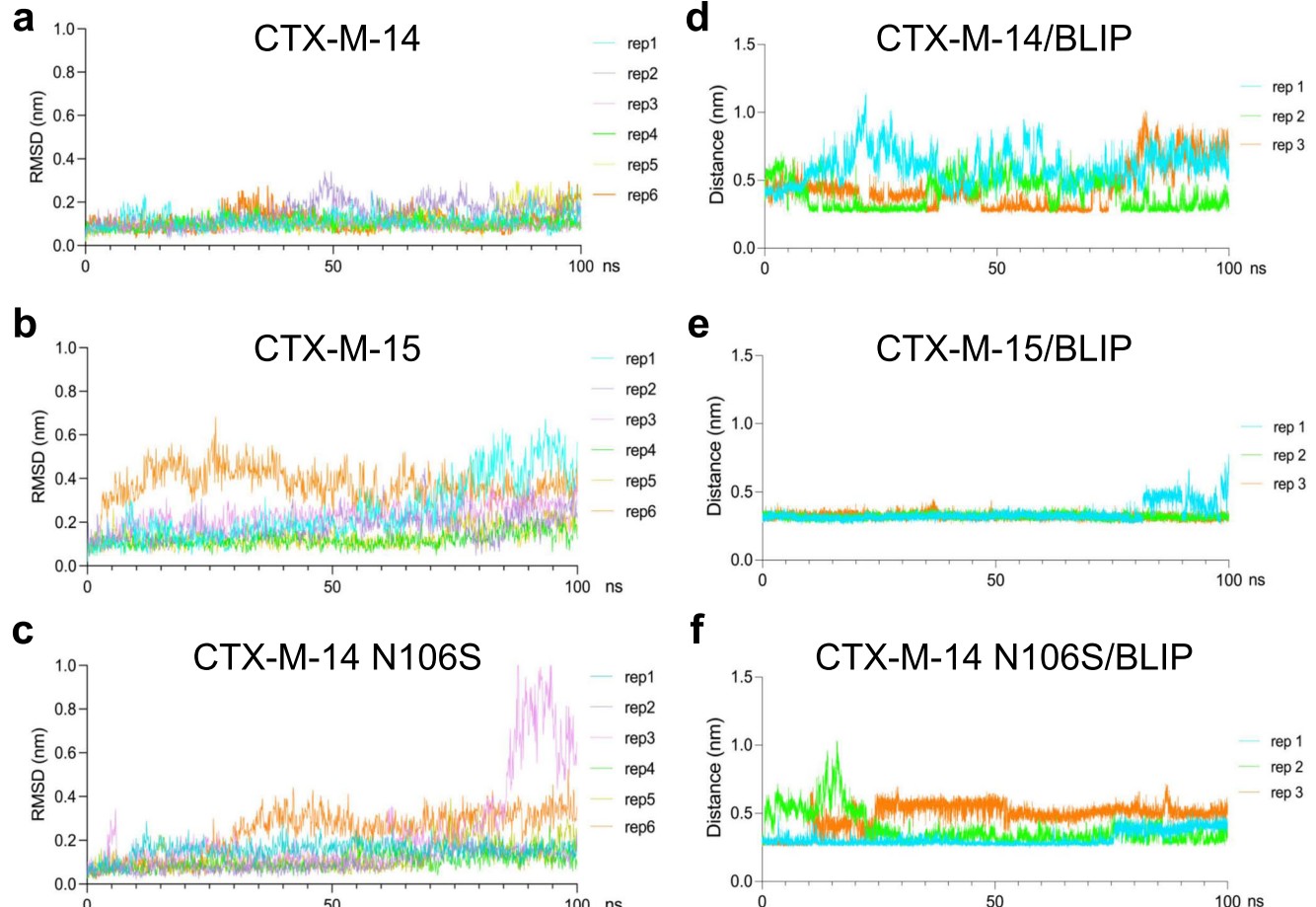

**Fig. 7 | Molecular dynamics simulations of CTX-M apo-enzymes and CTX-M/BLIP complexes. a** The RMSD of CTX-M-14 loop 103-106 backbone throughout the 100 ns MD simulations. Six replicate simulations are shown. **b** The RMSD of CTX-M-15 loop 103-106 backbone for six replicate 100 ns MD simulations. **c** The RMSD of CTX-M-14 N106S loop 103-106 backbone for six replicate 100 ns MD simulations. **d** The dynamic change of distance (nm) between CTX-M-14 Tyr105 backbone nitrogen and BLIP Glu73 side chain carboxylate oxygen atom throughout three replicate 100 ns MD simulations of a modeled structure of CTX-M-14 in complex with BLIP. **e** The dynamic change of distance (nm) between CTX-M-15 Tyr105 backbone nitrogen and BLIP Glu73 side chain carboxylate oxygen atom throughout three replicate 100 ns MD simulations using the CTX-M-15/BLIP X-ray structure. **f** The dynamic change of distance (nm) between CTX-M-14 N106S Tyr105 backbone nitrogen and BLIP Glu73 side chain carboxylate oxygen atom throughout three replicate 100 ns MD simulations of a modeled structure of CTX-M-14 N106S in complex with BLIP.

**Table 3 | B-factors for residues in the CTX-M-14 and CTX-M-15 β-lactamase 103-106 loop**

| Residue position | B-factor (Å²) | | | |
| --- | --- | --- | --- | --- |
| | main chain | side chain | all | normalized B-factor[a] |
| Val103 CTX-M-14 | 6.23 | 7.42 | 6.74 | −0.697 |
| Val103 CTX-M-15 | 8.34 | 9.27 | 8.74 | −0.200 |
| Asn104 CTX-M-14 | 7.65 | 10.80 | 9.22 | -0.082 |
| Asn104 CTX-M-15 | 8.23 | 16.64 | 12.44 | 0.304 |
| Tyr105 CTX-M-14 | 6.31 | 10.16 | 8.87 | −0.169 |
| Tyr105 CTX-M-15 | 9.79 | 25.68 | 20.38 | 1.386 |
| Asn106 CTX-M-14 | 6.83 | 7.27 | 7.05 | −0.620 |
| Asn106 CTX-M-15 | 8.78 | 16.18 | 12.48 | 0.309 |

[a]Normalized B factor calculated as $(B - <B>)/\sigma$, where $<B>$ is the average B-factor for the main chain and side chains for the structure and $\sigma$ is the standard deviation (SD). Average B-factor and SD for all (main chain and side chain) for the CTX-M-15 structure is 10.21 and 7.34, respectively. The avg B-factor and SD for CTX-M-14 is 9.55 and 4.03, respectively. The B-factors are calculated for CTX-M-15 PDB id:4HBT and CTX-M-14 PDB id:1YLT.

constructed by aligning the CTX-M-14 and CTX-M-14 N106S crystal structures with CTX-M-15 in the CTX-M-15/BLIP structure. The simulations showed that the key difference between the weak affinity CTX-M-14/BLIP and high-affinity CTX-M-14 N106S/BLIP and CTX-M-15/BLIP complexes is the hydrogen bond interaction of the side chain oxygen of BLIP residue Glu73 with the main chain NH groups of Tyr105, which are present during the majority of the simulations for CTX-M-14 N106S/BLIP and CTX-M-15/BLIP (Fig. 7e, f). The hydrogen bond with the NH of Tyr105 is made possible due to the flipping of the peptide bond between Asn104 and Tyr105 in CTX-M-15 and CTX-M-14 N106S

(Fig. 4c, e). In contrast, the BLIP Glu73 hydrogen bond to the NH of Tyr105 of CTX-M-14 is restricted due to the orientation of the Asn104-Tyr105 peptide bond (Fig. 7d). In addition, the interaction of the side chain of the BLIP hotspot residue Trp112 and the 103-106 loop is stable during the simulations of BLIP with CTX-M-14 N106S and CTX-M-15 but not in the CTX-M-14/BLIP simulation. Thus, the interactions with BLIP appear to be more stable with the CTX-M-15 complex. Taken together, the molecular dynamics simulations suggest that the differences in BLIP binding to CTX-M-14 and CTX-M-15 may be due to stability differences between the 103-106 loop in the two apo-enzymes as well as energetic differences within the two BLIP complexes.

## Discussion

The structure of BLIP in complex with the CTX-M-15 β-lactamase represents the first structure of an enzyme from the CTX-M group determined in complex with BLIP. The structure reveals a change in conformation of two loops in the CTX-M-15 active site when bound to BLIP compared to the apo-enzyme structure. Interestingly, both conformational changes are associated with a flip in the orientation of main chain peptide bonds. The CTX-M-15 103–106 loop, which is part of the larger protruding loop region, undergoes a conformational change upon BLIP binding whereby it closely resembles the loop in the high-affinity TEM-1/BLIP structure, suggesting this loop conformation is optimal for BLIP binding. In addition, the Gly213-Ser220 active site loop is in a different conformation in the CTX-M-15/BLIP complex compared to the apo-enzyme structure. In the altered conformation, residue Thr216 is positioned to make a hydrogen bond with residue Asp49 in the BLIP 45-52 loop that is present in the active site, suggesting the change in conformation enhances BLIP binding (Supplementary Fig. 5).

We have previously shown that the N106S substitution, which is found in CTX-M natural variants, increases the thermodynamic stability of the enzyme and also influences the substrate specificity of the enzyme for β-lactam antibiotics in that it lowers the catalytic efficiency for hydrolysis of the extended-spectrum cephalosporins cefotaxime and ceftazidime[13]. Because the N106S substitution alters the conformation of the 103–106 loop, we concluded that the conformation of the loop is an important determinant of extended-spectrum cephalosporin hydrolysis[13]. The structure of the CTX-M-14 N106S enzyme in complex with cefotaxime showed that the altered substrate specificity results from Asn104 rotating away from the active site in the altered loop conformation and disrupting a key hydrogen bond with the drug[13]. However, as shown here, these same conformational changes of the 103–106 loop induced by the N106S substitution enhance BLIP binding. Thus, the conformation of the 103–106 active site loop is an important determinant of both substrate specificity and BLIP inhibition potency for class A β-lactamases.

The finding that the 103–106 loop is present in the CTX-M-like conformation in the apo-enzyme structure of CTX-M-15 but in the TEM-like conformation in the CTX-M-15/BLIP complex indicates the enzyme can sample both conformations. It is not known if BLIP induces the change or if BLIP binds and stabilizes CTX-M-15 enzymes that sample the altered conformation of 103–106 loop in solution and thereby shifts the equilibrium towards the altered conformation. The N106S substitution clearly stabilizes the TEM-like conformation of the loop since this conformation is found in the X-ray structure of the CTX-M-14 N106S mutant apo-enzyme[13]. We hypothesize that the loop is present in both conformations in solution with the CTX-M-like structure being dominant and that either the N106S mutation or BLIP binding shifts the equilibrium towards the TEM-like conformation. By this view, CTX-M-14 is weakly inhibited by BLIP because it rarely samples the TEM-like conformation.

For the CTX-M-15/BLIP structure, there is also a change in the rotamer conformation of the Tyr105 phenoxy group coincident with the change in orientation of the 104–105 peptide bond of CTX-M-15

such that the Tyr105 side chain is angled down the active site and makes a hydrogen bond with the BLIP mainchain N of Gly48 and hydrophobic interactions with Ala47 (Supplementary Fig. 2). A similar change in Tyr105 rotamer conformation has been observed in all BLIP/β-lactamase complexes determined to date, including structures of BLIP with TEM-1, SHV-1, and KPC-2[24,25,35]. The altered rotamer conformation of Tyr105, however, is not strictly associated with the peptide bond conformation of residues 104–105 in that structures of TEM-1, SHV-1, and KPC-2 apo-enzymes do not show the altered rotamer at Tyr105, despite containing the same 104–105 peptide bond conformation as that found in the BLIP/β-lactamase complexes[36–39]. Note, however, that molecular dynamics studies on TEM-1 have shown that in the apo-enzyme both rotamer conformations of Tyr105 are significantly populated[40]. In addition, in our structure of the apo CTX-M-14 N106S enzyme, electron density is present for both rotamer conformations of Tyr105 and is associated with the change in orientation of the 104–105 peptide bond[13]. Therefore, the change in orientation of the 104–105 peptide bond may stabilize and thus increase the occupancy of the altered rotamer conformation of Tyr105 observed in all of the BLIP/β-lactamase structures.

Although the conformation of the 103–106 loop is clearly important for BLIP binding and inhibition of CTX-M β-lactamases, other interactions have also been shown to alter BLIP binding specificity with class A β-lactamases. For example, the TEM-1 and SHV-1 β-lactamases are 67% identical in sequence and have an identical conformation of the 103–106 loop and yet BLIP inhibits SHV-1 with ~1000-fold less potency than TEM-1. It has been shown that the presence of Glu104 in TEM-1 versus Asp104 in SHV-1 is an important component for the difference in affinity. The introduction of the D104E substitution into SHV-1 restores a salt-bridge between Glu104 in β-lactamase and Lys74 in BLIP and results in an ~1000-fold increase in BLIP potency[24,41]. In these cases, the 103–106 loop of SHV-1 is in the TEM-like conformation, consistent with tight binding by BLIP. One interpretation is that the TEM-like 103–106 conformation is a prerequisite for tight binding but that other suboptimal interactions can decrease binding affinity. Further, a Y50A substitution on the BLIP His45-Tyr51 loop, which does not directly contact the β-lactamase 103–106 loop, results in a 10-fold increase in BLIP inhibition potency for TEM-1 but a 15-fold decrease in potency for inhibition of the SME-1 class A β-lactamase[20]. However, structures of the BLIP Y50A complexes with β-lactamases are not available to assess conformational changes. Taken together, these findings suggest there are multiple components in the BLIP/β-lactamase interaction interface critical for high binding affinity.

BLIP has been suggested as a possible therapeutic modality due to its high potency for inhibition of β-lactamases[42]. Permeation of BLIP through the outer membrane of Gram-negative bacteria, however, represents a challenge for its therapeutic use as an inhibitor. Nevertheless, the results presented here and in previous reports indicate that BLIP can be readily engineered for altered β-lactamase binding specificity and affinity, which is advantageous for designing highly potent inhibitors[24,29,43]. Identifying the important interaction hotspots and the attendant conformational features that are important to the very strong interaction of BLIP with β-lactamases may also help guide the design of small-molecule inhibitors to combat drug resistance.

## Methods

### Protein expression and purification

Plasmid pET28a-CTX-M-14 used is this study was described previously[44]. The pET28a-CTX-M-15 plasmid was constructed by PCR amplification of the CTX-M-15 gene without the N-terminal signal sequence by colony PCR using the CTX-M-15 positive *Klebsiella pneumoniae* KPN11 strain as the template[45]. The forward and reverse primer sequences were 5′-GGCGGCCATATGCAAACGGCGGACGTAC-3′ and 5′-GGCGGCGAGCTCTTACAAACCGTCGGTGAC-3′, respectively. The amplified DNA fragment of CTX-M-15 was digested with the *Nde*I and

*Sac*I restriction endonucleases and the product was ligated into the pET28a plasmid. The sequence of the pET28-CTX-M-15 plasmid was confirmed by DNA sequencing. Site-directed mutagenesis was used to introduce N106S substitution into the pET28a-CTX-M-15 plasmid to create CTX-M-15 N106S. The forward primer 5′-CTT GTT AAC TAT AGC CCG ATT GCG GAA AAG-3′ and reverse primer 5′-C CGC AAT CGG GCT ATA GTT AAC AAG GTC AG-3′ were used for mutagenesis. The pET28-CTX-M-14 N106S mutant was constructed previously[13]. The CTX-M-15 V133T mutant was constructed by site-directed mutagenesis using the forward primer 5′-GGCCCGGCTAGCACCACCGCGTTCGCCCGACAGC TGGGA and reverse primer 5′-GGCGAACGCGGTGGTGCTAGCCGG GCCGCCAACGTGAGC. The QuikChange mutagenesis protocol[46] was performed using the Phusion DNA polymerase (ThermoFisher Scientific) for site-directed mutagenesis.

CTX-M-14, CTX-M-15 and the N106S variants of each enzyme were expressed from pET28 with an N- terminal His6 tag and purified from *E. coli* BL21(DE3) strains as described previously[47]. *E. coli* cells containing the β-lactamase expression plasmid were cultured in LB medium with 25 µg/ml kanamycin at 37 °C until OD reached 1.0–1.2. β-lactamase expression was induced by 0.2 mM isopropyl β-D-1-thiogalactopyranoside for 20 h at 23 °C. The following day, *E. coli* cells were harvested by centrifugation and resuspended in buffer A, consisting of 25 mM sodium phosphate (pH 7.4), 300 mM NaCl, and Xpert protease inhibitor mixture (GenDE- POT, Katy, TX, USA) supplemented with 20 mM imidazole. The cells were disrupted by sonication and insoluble material was removed by centrifugation at 8,000 g for 15 min. The supernatant was loaded onto a Co$^{2+}$ Talon resin column (Takara Bio USA, Inc., Mountain View, CA, USA). After washing, protein was eluted with buffer A supplemented with 40, 60, 80 and 100 mM imidazole, respectively. The elution fractions containing the expressed protein were then combined, concentrated, and buffer exchange was performed with buffer A using an Amicon® Ultra-15 centrifugal filter unit (MilliporeSigma, Burlington, MA, USA). The N-terminal His-tag of the purified protein was cleaved with tobacco etch virus (TEV) protease in an overnight reaction at 4 °C. The CTX-M enzymes with the His-tag removed were purified by incubating with a Ni$^{2+}$-charged-Sepharose 6 Fast Flow resin (GE Healthcare Life Sciences) for 1 h, 4 °C. Protein purity, typically 95%, and His-tag removal were analyzed by SDS-PAGE followed by Coomassie Brilliant Blue staining.

Wild-type BLIP was expressed from the pGR32 plasmid[21] and contained an N-terminal 6-His-tag. BLIP was expressed and purified from the *E.coli* RB791 strain as described previously[48]. *E. coli* cells carrying pGR32 plasmid were cultured in LB medium at 37 °C and induced by 6 mM D-lactose for 26 h at 23 °C when OD reached 0.9-1.0. Cells were harvested by low speed centrifugation and resuspended in buffer A containing 20 mM Tris-HCl (pH 8.0), 500 mM NaCl and Xpert protease inhibitor cocktail (GenDEPOT, Katy, TX). Sonication was used to disrupt cells and cell debris were removed by centrifugation at 8,000 g for 20 min. The supernatant was loaded onto a Co$^{2+}$-Charged Talon column and the flow through was collected and re-applied to the column to increase protein binding. After washing, BLIP protein was eluted by adding buffer A supplemented with 400 mM imidazole. To further purify BLIP, eluted protein fractions were concentrated and loaded onto a gel filtration column equilibrated with 20 mM Tris-HCl (pH 8.0), 200 mM NaCl buffer. Protein purity was visualized by SDS-PAGE followed by Coomassie Brilliant Blue (CBB) 160 staining.

### $K_i$ determinations for BLIP inhibition of β-lactamases

The inhibition constant for BLIP with CTX-M β-lactamases was determined with purified BLIP and β-lactamases. The $K_i$ was determined by incubating increasing concentrations of BLIP with β-lactamase for 20 min and subsequently adding the chromogenic β-lactam substrate nitrocefin and measuring the initial velocity of hydrolysis of nitrocefin at 482 nm. The reactions were performed in 50 mM sodium phosphate (pH 7.0) and 100 µg/ml bovine serum albumin (BSA). The initial

velocities of nitrocefin hydrolysis at various concentrations of BLIP were normalized by dividing by the hydrolysis rate in the absence of BLIP. The values were plotted versus BLIP concentrations used for each experiment and the data was fit using the Morrison equation for tight binding inhibitors as previously described to obtain the $K_i$[22,48,49] using GraphPad Prism v9.0 software. For BLIP inhibition of CTX-M-15, the reactions contained 0.05 nM of β-lactamase, 25 µM of nitrocefin and a $K_M$ value of 35 µM for nitrocefin hydrolysis by CTX-M-15 was used in the Morrison equation. For BLIP inhibition of CTX-M-14, the reactions contained 0.3 nM of β-lactamase, 50 µM of nitrocefin and a $K_M$ value of 25 µM for nitrocefin hydrolysis by CTX-M-14 was used in the Morrison equation. For BLIP inhibition of CTX-M-15 N106S, the reactions contained 0.2 nM of β-lactamase, 22.5 µM of nitrocefin and a $K_M$ value of 95 µM for nitrocefin hydrolysis by CTX-M-15 N106S enzyme was used in the Morrison equation. For BLIP inhibition of CTX-M-14 N106S, the reactions contained 0.6 nM of β-lactamase, 15 µM of nitrocefin and a $K_M$ value of 31 µM for nitrocefin hydrolysis by CTX-M-14 N106S enzyme was used in the Morrison equation. For BLIP inhibition of CTX-M-15 G240D, the reactions contained 0.5 nM of β-lactamase, 50 µM of nitrocefin and a $K_M$ value of 25 µM was used in the Morrison equation. For BLIP inhibition of CTX-M-15 V133T, the reactions contained 0.05 nM of β-lactamase, 50 µM of nitrocefin and a $K_M$ value of 25 µM was used in the Morrison equation.

### Differential scanning fluorimetry assay

The thermal shift assay was employed to measure the melting temperature ($T_m$) of the CTX-M-14 and CTX-M-15 β-lactamases. Protein denaturation was monitored via an increase in fluorescence of SYPRO Orange dye (ThermoFisher Scientific, USA) which binds to hydrophobic residues that get exposed as the protein unfolds. Briefly, purified CTX-M-14 and CTX-M-15 proteins were appropriately diluted in a buffer containing 50 mM Tris-HCl, pH 7.5, 150 mM NaCl. The assay was set up on a 384-well Roche plate where the CTX-M-14 and CTX-M-15 proteins were at a concentration of 2 µg, and SYPRO Orange dye at 5X in a 10-µL reaction. Thermal scanning (20 to 95 °C) was performed using a Roche Lightcycler 480 real-time PCR instrument and fluorescence intensity was measured after every 10 seconds. The data analysis was run on a Roche Lightcycler 480 real-time PCR instrument as well as LightCycler thermal shift analysis software. The graphs showing the denaturation versus temperature were generated using GraphPad Prism software. The data was averaged from three technical replicates.

### Protein crystallization

Purified CTX-M-15 and BLIP were mixed in a 1:1 molar ratio and incubated at 4 °C overnight to facilitate complex formation. The protein mixture was then concentrated to approximately 10 mg/ml. Crystal screening was performed using commercially available crystallization screens PEGs and PACT from Qiagen (Valencia, CA, USA) using the hanging-drop vapor diffusion method. Crystallization screens were set up using an in-house TTP LabTech Mosquito instrument (TTP Labtech Ltd., Melbourn, UK). Crystals were obtained in 20% w/v PEG 3350, 0.2 M NaF, 0.1 M Bis-Tris-propane pH 6.5.

### Crystallography data processing and refinement

Diffraction data were collected at the Berkeley Center for Structural Biology using the Advanced Light Source synchrotron 8.2.1 beamline. Reflection data were indexed, integrated, and scaled using the HKL2000[50] v722 and iMosflm[51] v7.4 in the CCP4i Suite v7.1[52]. Molecular replacement was performed using the CTX-M-15 apo structure (PDB id:4HBT) and the BLIP structure (PDB id:3GMU) with Phaser[53] v2.1.2. Structures were refined further for several iterative rounds with Phenix.refine v1.17.1 and Coot v1 density fitting[54–56]. Hydrogen bonding interactions were assessed with UCSF Chimera[57] v1.13.1, UCSF ChimeraX[58] v1.4, Coot[55], and Ligplot+ v2.2.4[59]. The amino acid

sequence alignment of CTX-M-14 and CTX-M-15 β-lactamases was performed using Clustal Omega[60] within the EMBL-EBI web server.

## Modeling and molecular dynamics simulations

The initial coordinates for all of the apo-CTX-M enzyme simulations come from the crystal structures downloaded from Protein Data Bank (PDB) entry (CTX-M-14 WT: 1YLT, CTX-M-14 N106S: 6CYK, CTX-M-15 WT: 4HBT). The initial coordinates for the CTX-M-15/BLIP complex are from the structure determined in this work, while CTX-M-14/BLIP and CTX-M-14 N106S/BLIP initial models were prepared by aligning CTX-M-14 and CTX-M-14 N106S crystal structures with CTX-M-15 using the PyMOL align command. The CHARMM-GUI website was used for PDB file clean-up, adding hydrogens, assigning protonation states (assuming a pH of 7.0 for the ionizable groups), and removing side chain ambiguity[61]. The protonation state of each histidine was checked taking into consideration their chemical environment.

GROMACS version 2020.6 and the CHARMM36-Jul21 forcefield were used for MD simulations[62,63]. For each set of initial coordinates, the periodic boundary condition was applied as follows: the box was set as cubic with the absolute size of each side greater than the largest dimension of the system by 10 Å. Explicit water was added using TIP3P water model, and the system charge was neutralized with sodium or chloride ions. A 50,000-step steepest descent energy minimization was performed to remove steric clashes or inappropriate geometry. Consecutive 100 ps NVT ensemble simulation with T at 300° K using modified Berendsen thermostat and 100 ps NPT ensemble simulations with T at 300 K and pressure at 1 bar were then performed to allow temperature and pressure coupling for protein and water-ion groups. The convergence of system temperature, pressure, and density was confirmed. Finally, the production run of 100 ns was performed with the time step of 2 fs. Short-range van der Waals cutoff distance was set at 1.2 nm. During the production run, coordinate frames were saved at every 10 ps. The trajectories were visualized through the following gmx trjconv commands: whole – center – mol(compact) – rot+trans. The CTX apo-enzyme simulations were repeated 6 times while the CTX-BLIP complex simulations were run in triplicate. The post-simulation distance and RMSD analysis after each run were performed by calling respective commands built in GROMACS. Loop 103-106 backbone atoms RMSD was calculated by least-square fitting with protein backbone atoms.

The MD simulation was performed using an in-house Linux cluster, which features two Dell Poweredge R740 servers, each with four 16-core Intel Xeon Gold 5218 processors and two NVIDIA Tesla V100S 32GB GPUs. The GROMACS GPU-enabled code was used and mainly run using GPU.

## Reporting summary

Further information on research design is available in the Nature Portfolio Reporting Summary linked to this article.

## Data availability

The BLIP/CTX-M-15 structure generated in this study has been deposited in the Protein Data Bank under accession code PDB id: 7S5S [https://doi.org/10.2210/pdb7S5S/pdb]. The structures used in this study can be found in the Protein Data Bank under the accession codes: 4HBT [https://doi.org/10.2210/pdb4HBT/pdb], 1YLT [https://doi.org/10.2210/pdb1YLT/pdb], 1JTG [https://doi.org/10.2210/pdb1JTG/pdb], 6CYK [https://doi.org/10.2210/pdb6CYK/pdb]. The enzyme inhibition source data and the differential scanning fluorimetry source data is available as a source data file. The command workflow for the molecular dynamics simulations is available as a supplementary note in the Supplementary Information file. The molecular dynamics trajectories have been deposited at Figshare [https://doi.org/10.6084/m9.figshare.21378981]. Plasmid expression constructs of BLIP and β-lactamases are available from the author upon request. Source data are provided with this paper.

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

## Acknowledgements

This work was funded by NIH grant AI32956 to T.P., Welch Foundation grant Q1279 to B.V.V.P., NIH grant CA259664 and Cancer Prevention and Research Institute of Texas grant RP220524 to M.P., and a start-up award from Texas Advanced Computing Center (TACC) to J.W. A.J. was funded by NIH training grant T32 GM120011. The ALS-ENABLE beamlines are supported in part by the National Institutes of Health, National Institute of General Medical Sciences, grant P30 GM124169-01. The Advanced Light Source is a Department of Energy Office of Science User Facility under Contract No. DE-AC02-05CH11231.

## Author contributions

S.L. performed experiments, analyzed data, and edited the manuscript. L.H. analyzed crystallography data, performed data visualization, and edited the manuscript. H.L. performed molecular dynamics simulations, data analysis, and edited the manuscript. A.J. performed biochemical experiments and edited the manuscript. P.R. performed biochemical experiments and edited the manuscript. M.P. performed the DSF stability experiments, analyzed data, performed data visualization and edited the manuscript. B.S. performed X-ray diffraction experiments and edited the manuscript. J.W. supervised molecular dynamics experiments and wrote manuscript. B.V.V.P. supervised structure determination, obtained funding, and wrote manuscript. T.P. supervised experiments, analyzed data, obtained funding, and wrote and edited the manuscript.

## Competing interests

J.W. is co-founder of Chemical Biology probes LLC and Coactigon Inc. The remaining authors declare no competing interests.
