## [Peer Review File · Nature Communications]

An Active Site Loop Toggles Between Conformations to Control Antibiotic Hydrolysis and Inhibition Potency for CTX-M β -lactamase Drug-Resistance EnzymesREVIEWER COMMENTS

Reviewer #1 (Remarks to the Author):

This is a very elegantly written manuscript describing key interactions between relevant residues of the CTX-M β -lactamases and BLIP. This study also includes the first structure of a CTX-M in complex with BLIP. Among the most striking results, the authors demonstrated that upon binding of the BLIP to the active site, the 103-106 loop undergoes a conformational change which seems to be crucial for the inhibition of the CTX-Ms, and in agreement with the remarkable inhibitory kinetic parameters.

I think that the evidence provided in this study will contribute to relevant advances in the design of novel therapeutic options, especially in the constant search for effective β -lactamase inhibitors.

The work clearly supports the conclusions, and the methodology is sound and neatly detailed.

I would only suggest keeping a constant nomenclature for the amino acids and the mutations: sometimes the one-letter code is used, and most others, the 3-letter one.

I strongly recommend accepting this manuscript for publication.

Reviewer #2 (Remarks to the Author):

The manuscript investigates the interactions between the CTX-M-14 and CTX-M-15 extended-spectrum β -lactamases and the β -lactamase inhibitory protein BLIP. The authors demonstrate BLIP to have by two orders of magnitude a greater affinity for CTX-M-15 than CTX-M-14. The crystal structure of the complex identifies that the loop bearing CTX-M-15 residues 103 – 106 undergoes a conformational change on BLIP binding to adopt an orientation resembling that observed in a CTX-M-14 N106S mutant. The N106S mutation was shown to enhance BLIP inhibition by two (CTX-M-14) and one (CTX-M-15) order(s) of magnitude. Molecular dynamics simulations indicate that the 103-106 loop samples more conformations in CTX-M-15 than in either CTX-M-14 or the CTX-M-14 N106S mutant. It is concluded that the relatively low affinity of CTX-M-14 for BLIP is due to infrequent sampling of the conformation of the 103-106 loop compatible with strong interactions with BLIP.

Although interactions of BLIP with other β -lactamases have been previously investigated, this is the first crystal structure for a CTX-M:BLIP complex. The data presented here add to a growing body of work (some of which is considered in the Discussion) describing how sequence variations affect BLIP affinity in different β -lactamases and β -lactamase classes. The experimental work has been carefully performed. The manuscript is relatively clearly written, although the descriptions of some of the specific interactions become quite involved and better use of superpositions and Figures might make the relevant conformational changes easier to visualize, particularly for the non-specialist. The main area for improvement would be the molecular dynamics simulations. These are of relatively short duration (100 ns) and have only been carried out once, and the rmsd changes that are identified are small (less than 1 Å) even for the system that is considered the most mobile (CTX-M-15). I would suggest that more robust data, derived from more extensive simulations with multiple replicates, are necessary to support the authors contention that there are differences in the dynamic behavior of the different systems.

Reviewer #3 (Remarks to the Author):

The manuscript describes some very interesting results concerning the different activities of beta-lactamase inhibitory protein (BLIP) towards two very similar proteins, CTX-M-15 and CTX-M-14. The biophysical and biochemical analysis reveals the crucial role of an active site loop conformation in CTX-M that contributes to both activity and inhibition. In addition to our understanding of the function and inhibition of beta-lactamases, an important group of antibacterial targets, these results will be of interest to researchers studying protein structure and function in general. The manuscript is concisely written. However, some revisions are needed, especially related to the origin of the BLIP's different activities towards CTX-M-14 and CTX-M-15.

1) Since the focus of the manuscript is the difference between CTX-M-14 and CTX-M-15, more information should be provided comparing these two enzymes. Are any of the sequence variations between the two proteins near the BLIP binding regions? For example, one difference between the two proteins is Gly240 (CTX-M-15) vs Asp240 (CTX-M-14). Residue 240 is in close contact with Phe142 from BLIP. BLIP Phe142 adopts different conformations in the complex with CTX-M-15 vs TEM-1, the latter of which has Glu240, somewhat similar to CTX-M-14. In the complex with CTX-M-14, BLIP Phe142 would be forced to assume the conformation similar to that in TEM-1 due to steric hinderance by Asp240. This Phe142 conformation is also in close contact with Asn104 and Tyr105 in the 103-106 loop. The flip of the peptide bond between Asn104 and Tyr105 is a key feature of the complex between CTX-M-15 and BLIP. Could residue 240 play a role in making the flip easier for CTX-M-15 than CTX-M-14 by allowing the alternative BLIP Phe142 conformation? Interestingly, for TEM-1 with Glu240, it also has Ser106, as the authors discussed. The manuscript shows that N106S mutation increases the potency of BLIP against CTX-M-14. These correlations suggest that more attention needs to be paid to residue 240 and its influence on BLIP Phe142.

2) In addition, if the 103-106 segment is more flexible in CTX-M-15 and may be involved in substrate specificity, is there any difference in the thermostability and beta-lactamase activity of CTX-M-15 and CTX-M-14?

3) The authors provided a global picture (Fig. 1) and some very detailed pictures (e.g., Fig. 3) of the interactions between CTX-M-15 and BLIP. It will be helpful to have an intermediate snapshot depicting all the BLIP residues interacting with the CTX-M-15 active site.

4) Fig. 2C can be somewhat confusing as it seems to suggest the K_i for CTX-M-14 was not accurately determined and only a range was given ($K_i \Rightarrow 100\text{nM}$). In the main text, the K_i was determined to be 800 nM. So some clarification in the figure legend could be helpful.

5) Page 18, Table 1, the unit for Resolution Range is missing. '1.4' should be '1.40'.

A footnote explaining the values in parentheses should be provided.

$I/\sigma(I)$ row is confusing. Not sure what the value in parentheses means.

Bond length and angle values are the rmsd values for these two parameters. But 'rmsd' is missing.

6) Page 20, Table 3, the unit for B factor is missing. Superscript 'a' is missing in the table and appears only in the footnotes.

Reviewer #4 (Remarks to the Author):

This is a well written analysis exploring clinically important β -lactamases (CTX-M -14 and -15) and structure activity relationships as revealed by BLIP binding

The authors determined the X-ray structure of

BLIP in complex with CTX-M-15 and comparison with apo-CTX-M-15 reveals that binding is associated with a conformational change of the protruding loop of β -lactamase, including a $\sim 180^\circ$ flip in orientation of the peptide backbone, suggesting the loop is in alternate conformations in solution or that BLIP induces the change. They found that the loop structure in

the complex is similar to that in a drug-resistant variant (N106S) of CTX-M-14 where the conformational change is associated with altered enzyme specificity for cephalosporin antibiotic hydrolysis. Since the N106S mutant loop conformation is similar to that in the CTX-M15/BLIP complex, the authors hypothesized that BLIP would bind N106S variants of CTX-M-14 and CTX-M-15 tighter since the pre-established favorable loop conformation would facilitate binding.

The authors found that the CTX-M N106S substitution results in ~ 100 - and 10 -fold increases in BLIP inhibition potency for CTX-M-14 and CTX-M-15, respectively. Thus, an active site loop in β -lactamase toggles between conformations that control antibiotic hydrolysis and inhibitor susceptibility. These findings highlight the role of accessible active site conformations in

controlling enzyme activity and inhibitor susceptibility as well as the influence of mutations in selectively stabilizing discrete conformations

RESPONSE TO REVIEWER COMMENTS

Reviewer #1 (Remarks to the Author):

This is a very elegantly written manuscript describing key interactions between relevant residues of the CTX-M β -lactamases and BLIP. This study also includes the first structure of a CTX-M in complex with BLIP. Among the most striking results, the authors demonstrated that upon binding of the BLIP to the active site, the 103-106 loop undergoes a conformational change which seems to be crucial for the inhibition of the CTX-Ms, and in agreement with the remarkable inhibitory kinetic parameters.

I think that the evidence provided in this study will contribute to relevant advances in the design of novel therapeutic options, especially in the constant search for effective β -lactamase inhibitors.

The work clearly supports the conclusions, and the methodology is sound and neatly detailed.

I would only suggest keeping a constant nomenclature for the amino acids and the mutations: sometimes the one-letter code is used, and most others, the 3-letter one.

I strongly recommend accepting this manuscript for publication.

Response: Thank you for the positive comments on the manuscript. We have revised to have a nomenclature of 3-letter in the text and one-letter in the body of figures for clarity. Mutants are labeled with standard nomenclature of N106S.

Reviewer #2 (Remarks to the Author):

The manuscript investigates the interactions between the CTX-M-14 and CTX-M-15 extended-spectrum β -lactamases and the β -lactamase inhibitory protein BLIP. The authors demonstrate BLIP to have by two orders of magnitude a greater affinity for CTX-M-15 than CTX-M-14. The crystal structure of the complex identifies that the loop bearing CTX-M-15 residues 103 – 106 undergoes a conformational change on BLIP binding to adopt an orientation resembling that observed in a CTX-M-14 N106S mutant. The N106S mutation was shown to enhance BLIP inhibition by two (CTX-M-14) and one (CTX-M-15) order(s) of magnitude. Molecular dynamics simulations indicate that the 103-106 loop samples more conformations in CTX-M-15 than in either CTX-M-14 or the CTX-M-14 N106S mutant. It is concluded that the relatively low affinity of CTX-M-14 for BLIP is due to infrequent sampling of the conformation of the 103-106 loop compatible with strong interactions with BLIP.

Although interactions of BLIP with other β -lactamases have been previously investigated, this is the first crystal structure for a CTX-M:BLIP complex. The data presented here add to a growing body of work (some of which is considered in the Discussion) describing how sequence variations affect BLIP affinity in different β -lactamases and β -lactamase

classes. The experimental work has been carefully performed. The manuscript is relatively clearly written, although the descriptions of some of the specific interactions become quite involved and better use of superpositions and Figures might make the relevant conformational changes easier to visualize, particularly for the non-specialist. The main area for improvement would be the molecular dynamics simulations. These are of relatively short duration (100 ns) and have only been carried out once, and the rmsd changes that are identified are small (less than 1 Å) even for the system that is considered the most mobile (CTX-M-15). I would suggest that more robust data, derived from more extensive simulations with multiple replicates, are necessary to support the authors contention that there are differences in the dynamic behavior of the different systems.

Response: Thank you for the comments on the manuscript. We have now completed 6 replicates of the 100 ns simulations for the RMSD of the CTX-M-14, CTX-M-15, and CTX-M-14 N106S apo-enzymes. The results are consistent, in that CTX-M-15 shows the larger differences in RMSD compared with CTX-M-14. Figure 7 has been revised to include the replicates.

We have also modified Figures 1 and 3 to provide panels with less detailed views to provide perspective and ease visualization of the conformational change. In addition, we have added a new figure (Fig. 5) that shows the conformational changes in a schematic to facilitate visualizing the conformational changes.

Reviewer #3 (Remarks to the Author):

The manuscript describes some very interesting results concerning the different activities of beta-lactamase inhibitory protein (BLIP) towards two very similar proteins, CTX-M-15 and CTX-M-14. The biophysical and biochemical analysis reveals the crucial role of an active site loop conformation in CTX-M that contributes to both activity and inhibition. In addition to our understanding of the function and inhibition of beta-lactamases, an important group of antibacterial targets, these results will be of interest to researchers studying protein structure and function in general. The manuscript is concisely written. However, some revisions are needed, especially related to the origin of the BLIP's different activities towards CTX-M-14 and CTX-M-15.

1) Since the focus of the manuscript is the difference between CTX-M-14 and CTX-M-15, more information should be provided comparing these two enzymes. Are any of the sequence variations between the two proteins near the BLIP binding regions? For example, one difference between the two proteins is Gly240 (CTX-M-15) vs Asp240 (CTX-M-14). Residue 240 is in close contact with Phe142 from BLIP. BLIP Phe142 adopts different conformations in the complex with CTX-M-15 vs TEM-1, the latter of which has Glu240, somewhat similar to CTX-M-14. In the complex with CTX-M-14, BLIP Phe142 would be forced to assume the conformation similar to that in TEM-1 due to steric hinderance by Asp240. This Phe142 conformation is also in close contact with Asn104 and Tyr105 in the

103-106 loop. The flip of the peptide bond between Asn104 and Tyr105 is a key feature of the complex between CTX-M-15 and BLIP. Could residue 240 play a role in making the flip easier for CTX-M-15 than CTX-M-14 by allowing the alternative BLIP Phe142 conformation? Interestingly, for TEM-1 with Glu240, it also has Ser106, as the authors discussed. The manuscript shows that N106S mutation increases the potency of BLIP against CTX-M-14. These correlations suggest that more attention needs to be paid to residue 240 and its influence on BLIP Phe142.

Response: This is a good point. In response, we constructed the G240D substitution in CTX-M-15 and determined the K_i for BLIP inhibition using purified protein. It was found that the K_i is the same as that for wt CTX-M-15, indicating this change is not responsible for the observed differences in potency. Along the same lines, CTX-M-14 contains Thr133, which hydrogen bonds with Asn106 and could affect loop conformation. This residue is Val133 in CTX-M-15. We constructed the V133T mutant in CTX-M-15, but again found no change in K_i for BLIP inhibition compared to wt CTX-M-15. Thus, these residue positions that are close to the active site do not seem to influence the conformational change or potency. These results have been added to the manuscript in the text and Figure 2.

2) In addition, if the 103-106 segment is more flexible in CTX-M-15 and may be involved in substrate specificity, is there any difference in the thermostability and beta-lactamase activity of CTX-M-15 and CTX-M-14?

Response: We measured the T_m of CTX-M-14 and CTX-M-15 using Differential Scanning Fluorimetry (DSF). The results are shown in Supplementary Fig. 4. Consistent with our hypothesis, CTX-M-15 is less stable than CTX-M-14.

3) The authors provided a global picture (Fig. 1) and some very detailed pictures (e.g., Fig. 3) of the interactions between CTX-M-15 and BLIP. It will be helpful to have an intermediate snapshot depicting all the BLIP residues interacting with the CTX-M-15 active site.

Response: We have added two panels to Fig. 1, providing an intermediate view and highlighting the key BLIP residues interacting with CTX-M-15 that can provide a frame of reference for the more detailed figures. In addition, two panels (A and B) have been added to Figure 3 providing an intermediate view of the active site in the same orientation as that shown in the more detailed panels C-H. Figure 5 has been moved and is now Figure 4 and a new Figure 5 shows a schematic of the conformational changes. The old Figure 4 is now Supplementary Fig. 3.

4) Fig. 2C can be somewhat confusing as it seems to suggest the K_i for CTX-M-14 was not accurately determined and only a range was given ($K_i > 100\text{nM}$). In the main text, the K_i was determined to be 800 nM. So some clarification in the figure legend could be helpful.

Response: The 800 nM K_i is a previously published value. We had confirmed this by showing the K_i was >100 nM. In response to the reviewer comment, we did a complete inhibition curve and determined a value for the K_i of 330 nM, which is somewhat lower than the published value but nevertheless >100 -fold higher than the K_i for CTX-M-15.

5) Page 18, Table 1, the unit for Resolution Range is missing. '1.4' should be '1.40'.

A footnote explaining the values in parentheses should be provided.

l/sig(l) row is confusing. Not sure what the value in parentheses means.

Bond length and angle values are the rmsd values for these two parameters. But 'rmsd' is missing.

Response: Table 1 has been corrected.

6) Page 20, Table 3, the unit for B factor is missing. Superscript 'a' is missing in the table and appears only in the footnotes.

Response: These items have been corrected in Table 3 with the superscript 'a' now present in both the table and footnotes. Also, note that we have normalized the B-factors according to accepted methods in the literature using the equation $(B - \langle B \rangle) / \sigma$, where $\langle B \rangle$ is the average B-factor for main chain and side chains for the structure and σ is the standard deviation (SD). This replaces the previous normalization done simply by dividing the B-factor by the average B-factor for the structure. This does not change any conclusions but uses a more accepted metric for normalization.

Reviewer #4 (Remarks to the Author):

This is a well written analysis exploring clinically important β -lactamases (CTX-M -14 and -15) and structure activity relationships as revealed by BLIP binding

The authors determined the X-ray structure of

BLIP in complex with CTX-M-15 and comparison with apo-CTX-M-15 reveals that binding is associated with a conformational change of the protruding loop of β -lactamase, including a $\sim 180^\circ$ flip in orientation of the peptide backbone, suggesting the loop is in alternate conformations in solution or that BLIP induces the change. They found that the loop structure in

the complex is similar to that in a drug-resistant variant (N106S) of CTX-M-14 where the conformational change is associated with altered enzyme specificity for cephalosporin antibiotic hydrolysis. Since the N106S mutant loop conformation is similar to that in the CTX-M15/BLIP complex, the authors hypothesized that BLIP would bind N106S variants of CTX-M-14 and CTXM-15 tighter since the pre-established favorable loop conformation would facilitate binding.

The authors found that the CTX-M N106S substitution results in ~100- and 10-fold increases in BLIP inhibition potency for CTX-M-14 and CTX-M-15, respectively. Thus, an active site loop in β -lactamase toggles between conformations that control antibiotic hydrolysis and inhibitor susceptibility. These findings highlight the role of accessible active site conformations in controlling enzyme activity and inhibitor susceptibility as well as the influence of mutations in selectively stabilizing discrete conformations

Response: We thank the reviewer for the comments on the manuscript.

REVIEWERS' COMMENTS

Reviewer #3 (Remarks to the Author):

The revision has addressed the comments from the previous review. The additional experiments have improved the rigor of the investigation. The manuscript is ready for publication.

RESPONSE TO REVIEWER COMMENTS

Reviewer #3 (Remarks to the Author):

The revision has addressed the comments from the previous review. The additional experiments have improved the rigor of the investigation. The manuscript is ready for publication.

Response: We thank the reviewer for the comments on the manuscript.